



# Climate benchmarks and input parameters representing locations in 68 countries for a stochastic weather generator, CLIGEN

Andrew T. Fullhart[1], Mark A. Nearing[1], Gerardo Armendariz[1], Mark A. Weltz[2]

[1]Southwest Watershed Research Center, USDA-ARS, 2000 E. Allen Rd., Tucson, AZ, 85719, U.S.A.
[2]Great Basin Rangelands Research Unit, USDA-ARS, 920 Valley Rd., Reno, NV, 89512, U.S.A.

*Correspondence to*: Andrew T. Fullhart (andrew.fullhart@usda.gov)

**Abstract.** This dataset contains input parameters for 12,703 locations around the world to parameterize a stochastic weather generator called CLIGEN. The parameters are essentially monthly statistics relating to daily precipitation, temperature and solar radiation. The dataset is separated into three sub-datasets differentiated by having monthly statistics determined from 30-year, 20-year, and 10-year minimum record lengths. Input parameters related to precipitation were calculated primarily from the NOAA GHCN-Daily network. The remaining input parameters were calculated from various sources including global meteorological and land-surface models that are informed by remote sensing and other methods. The new CLIGEN dataset includes inputs for locations in the U.S., which were compared to a selection of stations from an existing U.S. CLIGEN dataset representing 2,648 locations. This validation showed reasonable agreement between the two datasets, with the majority of parameters showing less than 20% discrepancy relative to the existing dataset. For the three new datasets, differentiated by the minimum record lengths used for calculations, the validation showed only a small increase in discrepancy going towards shorter record lengths, such that the average discrepancy for all parameters was greater by 5% for the 10-year dataset. The new CLIGEN dataset has the potential to improve the spatial coverage of analysis for a variety of CLIGEN applications, and reduce the effort needed in preparing climate inputs. The dataset is available at the National Agriculture Library Data Commons website at https://data.nal.usda.gov/dataset/international-climate-benchmarks-and-input-parameters-stochastic-weather-generator-cligen and https://doi.org/10.15482/USDA.ADC/1518706 (Fullhart et al., 2020c).





## 1 Introduction

Essential climate variables defined by the World Meteorological Organization are physical, chemical, or biological variables, or groups of linked variables that critically contribute to the characterization of Earth's climate (Bojinski et al., 2014). Aside from their use in climate studies, basic essential climate variables like precipitation and temperature are important for water resource management, drought monitoring, agricultural engineering, and other applications (Hollmann et al., 2013). The temporal resolution of climate data varies for these applications. Climate data reduced to monthly statistics may facilitate

analysis of multi-decadal climate trends and serve as benchmarks of climate normals (Menne et al., 2012; Hollmann et al., 2013). In this paper, it is discussed how a stochastic weather generator may be parameterized with a new dataset of monthly climate statistics to simulate daily weather outputs for locations around the world.

       Stochastic weather generators are used for a variety of applications that include model forcing, statistical downscaling of climate models, and study of climate change scenarios (Vaghefi and Yu, 2017). CLImate GENerator (CLIGEN) is one such

point-scale weather generator that produces daily outputs based on input parameters that are essentially observed monthly statistics. CLIGEN is regularly used to provide soil erosion models with realistic trends and statistical distributions of weather parameters (Kinnell 2019). Such models include the Rangeland Hydrology and Erosion Model (RHEM); the Water Erosion Prediction Project Model (WEPP); and the Revised Universal Soil Loss Equation 2 Model (RUSLE 2). CLIGEN can generate long-term realizations of stationary climate, subsequently enabling long-term erosion simulations, and ensuring that average

annual erosion rates reach convergence (Baffaut et al., 1996). CLIGEN has been validated in a number of countries, under a variety of climates, and for different outputs that include daily precipitation, peak intensity, time-to-peak intensity, storm duration, and storm frequency. For example, Mehan et al. (2017) showed that the mean of all daily precipitation values was within 0.1 mm of observations, and minimum and maximum daily temperatures within 0.1 °C for locations in the western Lake Erie basin. A particularly important CLIGEN output is precipitation intensity because of its high model sensitivity in

erosion and runoff modeling (Nearing et al., 2005). Zhang et al. (2008) validated intensity for the loess plateau of China based on distributions of maximum 30-min intensities ($I_{30}$) that were derived from CLIGEN's peak intensity. They found that differences with observed distributions were statistically insignificant, suggesting that rainfall erosivity could be accurately estimated using CLIGEN.

       CLIGEN has a dataset of location-specific input parameters for the United States with dense coverage, but on a global

scale, input parameters are sparsely available. This is partly because of the labor-intensive nature of determining the parameters, and because of numerous data requirements, e.g., high-frequency precipitation measurements. For erosion modeling, the lack of widely available CLIGEN inputs has hindered progress towards increasing the spatial scale and coverage of analysis that other aspects of soil erosion research have brought to the global scale, one example being the development of global maps of annual rainfall erosivity (Panagos et al., 2017). Hence, in the interest of increasing the availability of CLIGEN



inputs for soil erosion modeling and other applications, we present a dataset of CLIGEN input parameter files. The dataset represents 12,703 locations in 68 countries. Besides providing the necessary parameters to run CLIGEN simulations, the dataset also serves to provide statistics for representing climate normals. The parameters are validated using an existing CLIGEN input dataset for the United States, and differences are discussed.

## 2 Datasets

### 2.1 Overview

Three sets of CLIGEN v5.3 input files for international locations are presented, differentiated by having monthly parameters determined from minimums of 30-year, 20-year and 10-year records (note that assumptions were made to handle data gaps which are discussed in Sect. 2.2) (Fullhart et al., 2020c). The distribution of locations for the three datasets are in Fig. 1, which shows 7,673 parameter sets based on 30-year records (left panel), 2,336 parameter sets based on 20-year records 75 (middle panel), and 2,694 parameter sets based on 10-year records (right panel). All locations are unique, with no overlap in locations between the three datasets. As may be seen in Fig. 1, there is relatively sparse coverage for South America, Africa and southern Asia, while North America, Europe and Australia have relatively dense coverage. Table 1 enumerates the number of stations on each continent. Furthermore, a .kmz map layer is available on the Ag Data Commons website (link given in Sect. 4) that can be imported into Google Earth as an interactive map and allows the CLIGEN station nearest to an area of 80 interest to be found.

As 30 years is traditionally the minimum record length needed to represent climate, the 30-year dataset may be used to characterize climate normals (Bojinski et al., 2014). The 20-year and 10-year datasets, reflecting the most recent monthly records available at each location, may be more representative of current climates in some cases considering the non-stationarity of current and projected climate conditions (IPCC 2013). In soil erosion modeling, a 20-year record has been 85 suggested as the minimum length needed to represent rainfall erosivity (Wischmeier and Smith, 1978), which may be estimated using CLIGEN (Lobo et al., 2013). It should be noted that in non-stationary climates, CLIGEN inputs may be adjusted to represent departures from climate normals (Pruski and Nearing 2002; Zhang 2005; Vaghefi and Yu, 2016). For example, Zhang et al. (2013) determined how CLIGEN's precipitation intensity and skewness factors scale with monthly precipitation to correct for future changes in precipitation.

90 A list of parameters and their definitions that were determined for each input file are given in Table 2. These parameters are used to model statistical distributions that are randomly sampled by CLIGEN to derive daily outputs. Some parameters such as *TMAX AV* and *TMIN AV* are also typical climate benchmarks. Another climate benchmark, average monthly precipitation, may be determined by the following calculation from input parameters:

95 $$\text{avg. monthly precip.} = \text{n days} * MEAN\ P * \{P(W|D)\ /\ [1 - P(W|W) + P(W|D)]\} \tag{1}$$



where *n days* is the number of calendar days in the month being considered.

The various input parameters were derived from an assortment of data sources. In general, there were two main categories of sources: (1) ground-based precipitation networks, and (2) land-surface and meteorological models that assimilate remote sensing data and ground observations, and which reproduce historical time-series of variables of concern. The sources of data had various temporal resolutions. The data was used to make direct calculation of parameters, and for parameters where the available data was insufficient for direct calculation, parameter estimations were done. Each data source and the parameters are discussed in detail in the following sections.

## 2.2 Precipitation Accumulation

The primary source of precipitation data is the Global Historical Climate Network-Daily (GHCN-Daily) maintained by NOAA (Menne et al., 2012). The locations shown in Fig. 1 correspond to those of selected stations from GHCN-Daily. These ground-based records enabled direct calculation of five parameters related to precipitation accumulation: *MEAN P*, *S DEV P*, *SKEW P*, *P(W/W)* and *P(W/D)* (see Table 1 for their definitions). The GHCN-Daily dataset undergoes rigorous quality control, both to check for consistency of formatting, and for the integrity of daily values. Values are removed that fail any test in a suite of quality tests which identify a variety of problems. Durre et al. (2010) outlined 19 of the quality tests in detail.

Short record lengths and missing data precluded a wide majority (~90%) of GHCN-Daily stations from being used to create CLIGEN input parameters. A substantial number of data gaps necessitated an assumption for the calculation of the five monthly parameters related to accumulation. To handle gaps, records were queried starting with the most recent year available and going backwards in each time-series until the number of months needed could be produced by replacing gaps with existing records from earlier in the time-series. Therefore, it was assumed that time-series do not need to be temporally continuous. This means that records were accepted which did not necessarily come from sequential months, but which had at least 30, 20 and 10 complete individual months for each calendar month, in order to derive the 30-year, 20-year and 10-year monthly statistics, respectively. As a result, record lengths were queried that were often longer than the number of years needed. Also, since representing recent data was a priority, 96% of stations included at least some data after the year 2000, and 81% included some data after the year 2010. Ranges of years queried for each station are given in an extensive table available on the Ag Data Commons website (link given in Sect. 4). The ranges are defined by the first and last year with at least one monthly record accepted for use. Ranges in excess of the 30, 20 and 10-year minimum record lengths are due to data gaps for respective datasets.

## 2.3 Precipitation Intensity

In soil erosion and runoff modeling, precipitation intensity is a critical factor (Pruski and Nearing, 2002; Nearing et al., 2005). The two parameters related to precipitation intensity, *MX.5P* and *Time Pk*, require data with high frequency measurements such that hyetographs for a single precipitation event may be resolved. Since GHCN-Daily did not have





adequate temporal resolution, *MX.5P* was estimated from the daily data using a temporal downscaling model, and *Time Pk* was assumed to follow known average *Time Pk* values for given Köppen-Geiger climate classifications. The development of these procedures is discussed in Fullhart et al. (2020a) and Fullhart et al. (2020b). High resolution data needed for these procedures came from the Automated Surface Observing System (ASOS) maintained by NOAA with stations distributed across the United States and its territories (Doesken et al., 2002).

In CLIGEN, the *MX.5P* input parameter is used to parameterize statistical distributions of normalized peak intensity. The definition of *MX.5P* is as follows:

$$MX.5P = \frac{1}{k}\sum_{i=1}^{n=k} maxI_{30_i} + \cdots + maxI_{30_n} \tag{2}$$

where $k$ is the number of times (years) a record for a given month exists in the data set, and *maxI*$_{30}$ is the maximum 30-minute intensity (mm hr$^{-1}$) for each monthly record (Yu 2005). Since maximum 30-minute intensity is most accurately determined from data with as high frequency of measurement as possible, deriving values from data with lower resolutions results in underestimation bias, therefore necessitating use of the temporal downscaling model for *MX.5P*. The downscaling model took GHCN-Daily data to estimate the *MX.5P* value that would be expected if derived from the 1-min data. The downscaling model is a machine learning regression using Gradient Boosting trained with 609 ASOS stations (Fullhart et al., 2020b). The model requires 11 predictor variables shown in Table 3, which are statistics that may be determined from daily data and geographic information, some of which are already CLIGEN inputs. While *MX.5P* from 1-min resolution was estimated by the model, the predictor variable with the single most predictive power was *MX.5P* derived from daily data, which was calculated based on an assumption that intensity was constant for the duration of daily intervals (and was therefore grossly underestimated). *MEAN P* and *S DEV P* were also important predictors. The *MX.5P* values estimated by the model were found to have an RMSE of 0.148 inches (3.76 mm) (Fullhart et al., 2020b).

The second intensity parameter, *Time Pk*, represents values at 12 equal intervals along the probability density function of normalized time-to-peak intensity for events recorded at a given station (*Time Pk* is the only input parameter that does not represent monthly values, though there are 12 values per station, each representing quantiles of the PDF). For a given *Time Pk* interval, the definition is as follows:

$$Time\ Pk(i) = \frac{N_{tp(i)}}{N_{tot}} \tag{3}$$

where *Time Pk*($i$) is the *Time Pk* value at interval $i$; $tp$ is time-to-peak intensity normalized to the event duration; $N_{tp(i)}$ is the number of events where $tp <= i$; and $N_{tot}$ is the total number of events. Interval, $i$, ranges between 1/12 and 12/12, and varies by increments of 1/12. (Yu 2005). Events were separated by $>= 6$ hours of no precipitation.

In Fullhart et al. (2020a), it was shown that using climate average *Time Pk* values for the Köppen-Geiger climate classification of a given station resulted in <10% error relative to true *Time Pk* values, suggesting little variation of *Time Pk*





within climate classifications. In this previous study, a different weather station network was used—the U.S. Climate Reference Network (USCRN) at 5-min resolution (Diamond et al., 2013). For the new dataset of CLIGEN inputs, the analysis was repeated for the climate classifications represented by the 1-min ASOS network, though in some cases, climate classifications exclusive to the USCRN were used. Table A1 shows the assumed *Time Pk* values for each climate classification. Of the 30 highest-order climate classifications, 19 were represented by ASOS and USCRN. The remaining 11 classifications were assumed to be the averages of the other *Time Pk* values within respective first-order groups (of which there are 5, where A is tropical, B is arid, C is temperate, D is cold, and E is polar). As such, the climate classification of each station was used to index the assumed *Time Pk* values used in the CLIGEN input files. The climate classification of each station was determined based on the Köppen-Geiger climate map of Beck et al. (2018) representing the 1980-2016 time period at 0.083° resolution.

### 2.4 Temperature

The 5 temperature-related parameters, *TMAX AV*, *TMIN AV*, *SD TMAX*, *SD TMIN* and DEW PT, have straightforward calculations. However, the required data were only available for a subset of GHCN-Daily stations. To avoid limiting the analysis to this subset of stations, these data were instead derived from the model outputs of the ERA5 global meteorological/climate analysis ("ECMWF ReAnalysis", with ERA5 being the fifth major global reanalysis). The ERA5 analysis was created by The European Centre for Medium-Range Weather Forecasts and the Copernicus Climate Change Service (Balsamo et al., 2018). ERA5 provides climate and land-surface outputs at various temporal resolutions, including daily and monthly. Google Earth Engine was used to download maximum and minimum temperatures at daily resolution, and average dew point temperatures at monthly resolution. Use of the ERA 5 model also allowed continuous time-series to be obtained without gaps for the 30-year, 20-year and 10-year datasets (from 1990 through 2019, 2000 through 2019, and 2010 through 2019, respectively).

### 2.5 Solar Radiation

Incoming shortwave radiation is represented in CLIGEN by the *SOL.RAD* and *SD RAD* parameters which require that daily solar radiation is known with units of langley/d where 1 langley = 41,840 J/m². These parameters were calculated with relatively high frequency (3-hr) measurements that captured daily and day-to-day variability of radiation. This data came from the Global Land Data Assimilation System model (GLDAS) produced by NASA at averaged 3-hr intervals (Fang et al., 2009). The outputs of the reprocessed GLDAS 2.0 and GLDAS 2.1 versions were used and download from Google Earth Engine. The most recent data available was used to create continuous time-series with temporal ranges being the same as those for the temperature parameters. For an individual day, incoming solar radiation was modeled by fitting a gaussian curve through the 3-hr time-averaged data points. Doing this avoided underestimation caused by time-averaging, which would have occurred by considering the 3-hr datapoints alone. Also, if the 3-hr intervals did not coincide with the time of peak intensity, comparison to Ameriflux data (discussed more later) showed that the gaussian curve tended to better approximate peak radiation than the greatest 3-hr datapoint.



A number of stations that existed on coasts or on small islands, particularly in the Pacific Ocean, did not have solar
radiation data coverage for their locations because the GLDAS product covers only locations beyond a certain coastal
proximity. In total, 390 stations had this problem. For these stations, data from the nearest station with existing data was used.
300 of the stations with missing data were within 100 km of a station with data. Some proximities, however, were much further,
with islands in the south Pacific being examples. Similarly, some locations in the existing U.S. CLIGEN input dataset used for
validation (Srivastava et al., 2019) did not have observed solar radiation, and their parameter values were taken from the
nearest station with available data, which in some cases were at considerable distances, potentially leading to poor validation
in Sect. 3.

To ensure locations are matched for validation, a separate validation from that of Sect. 3 was done for solar radiation
parameters. In this, GLDAS output was compared to 10 ground-based Ameriflux stations that monitor ecosystem fluxes
including solar radiation (Hargrove et al., 2003). The Ameriflux network has stations distributed across the North and South
American continents, and the 10 stations were selected from a range of latitudes and climates as a representation of global
variability. From these stations, a single year was selected that had the fewest data gaps. Comparison to corresponding GLDAS
outputs showed reasonable agreement with an RMSE of 36.6 langley/d and with GLDAS being overestimated by <1% for
monthly values of *SOL.RAD*. Error was more evident for *SD RAD* suggesting that GLDAS was not optimum for capturing the
day-to-day variability of radiation. The RMSE for *SD RAD* was 38.6 langley/d with GLDAS being underestimated by 24.1%.

## 2.5 Wind

Very few applications of CLIGEN have used wind data in the past, perhaps the only one being the blowing snow
component in WEPP (Nicks et al., 1989). CLIGEN inputs require high-frequency measurement of wind speed (m/s) and
azimuthal wind direction. This includes mean, standard deviation, and skewness of daily wind speed on a monthly basis; and
determinations of the average daily percentage of time with wind directions coming from the 4 cardinal directions, 4
intercardinal directions, and the 8 sub-divisions of these (e.g. NNE, ENE), on a monthly basis. However, wind data was not
obtainable for the locations corresponding to the GHCN-Daily stations with the level of detail needed for creating CLIGEN
input files. The solution to this was to use the "International Conversion Programs" tool (availability given in Sect. 4), which
takes the known daily precipitation accumulation and temperature parameters from an international station of interest and finds
the existing station in the U.S. CLIGEN dataset with the most similar climate, allowing its wind parameters to be used (and
other remaining parameters, if needed). Information regarding the locations from where wind parameters were taken from are
given at the bottom of each input file.

## 3 Validation

Each parameter except for the wind parameters were compared to an existing dataset for the U.S. and its territories
created in 2015 using NOAA NCDC DSI-3260 data at 15-min resolution and consisting of 40-year records for 2,648 stations



225 (Srivastava et al., 2019). This limited the validation to only stations for the U.S., and from those, only the new stations within 10 km of an existing CLIGEN station were accepted. This resulted in the validation of 61 stations for the 30-year dataset, 53 stations for the 20-year dataset, and 204 stations for the 10-year dataset. For each of the validated parameters, RMSE, percent bias, and percent error were determined, where it was assumed that values from the existing U.S. dataset were the true values (performance metric definitions are given in Table A2). A summary of the validation is seen in Table 4. Inconsistencies 230 between the two datasets were attributed to: differences of data sources, differences in temporal resolution of data used, differences in record lengths, and whether data was interpolated or taken from nearby stations.

Overall, reasonable agreement was found, with PERROR being below 20% for the majority of parameters. As expected, record length is a factor in the comparison to the 40-year U.S. dataset. Percent error increased slightly on average (~5%) with decreasing record length, going from the 30-year to 10-year dataset. Though a small increase, this difference likely 235 reflected the potential for capturing short-term climate dynamics by the 20-year and 10-year datasets. For the 5 parameters related to daily accumulation, the parameter with the highest error was *SKEW P*, with error up to 30%. The sign of PBIAS for *SKEW P* was consistently positive suggesting that the GHCN-Daily data showed less skewness towards high daily accumulation.

Error was also considerable for the two parameters related to precipitation intensity, *MX.5P* and *Time Pk*. The 240 discrepancies were due to multiple issues including the fact that the DSI-3260 dataset uses 15-min resolution compared to the 1-min resolution that the *MX.5P* downscaling model and *Time Pk* distributions were based on. As mentioned, the downscaling model was previously shown to produce an average error of 0.148 inches (3.76 mm) (Fullhart et al., 2020b). In the comparison to the DSI-3260 dataset, downscaled *MX.5P* values resulted in discrepancy of up to 37% error for *MX.5P*. Interval values for *Time Pk* distributions were generally smaller in magnitude and approached unity later in the distribution, meaning that the 245 peak intensity of storms generally happened later in their duration than in the DSI-3260 data. This may be expected given the relatively coarse 15-min resolution of DSI-3260, and particularly when considering shorter storms, such as convective storms, the apparent peak intensity may have considerable uncertainty.

Temperature parameters were generally in agreement with no consistent estimation bias, except for *DEW PT*, which was slightly underestimated on average by up to 6%. Errors for *SOL.RAD* were up to 6%, with a slight overestimation bias of 250 up to 3%. While *SOL.RAD* was in good agreement, *SD SOL* indicated up to 193% more day-to-day variability of solar radiation. The GLDAS data for solar radiation generally agreed better with the variability of the Ameriflux network that was discussed in Sect. 2.5, with GLDAS showing 24% less variability than Ameriflux. Given the reasonable agreement between GLDAS and Ameriflux, and good agreement of *SOL.RAD* with the DSI-3260 data, the substantial underestimation bias of *SD SOL* may be the result of errors in the existing U.S. inputs.

While the U.S. represents a wide range of climate types, limitation of the validation to only the U.S. is a hinderance to quality assurance of the new dataset. However, each of the source data have their own quality assurances prior to going to product. Particularly for the ERA5 and GLDAS global products, biases are documented and are known to happen on regional and continental spatial scales, and may relate to extremes in temperature, moisture, geographic location, etc. (Zhou et al., 2013;

Ji et al., 2015; Urraca et al., 2018; Wang et al., 2019). Therefore, the uncertainty of each CLIGEN parameter also depends on
the particular source data.

## 4 Data Availability

The new international CLIGEN dataset is available at the National Agriculture Library website—Ag Data Commons—at https://data.nal.usda.gov/dataset/international-climate-benchmarks-and-input-parameters-stochastic-weather-generator-cligen (Fullhart et al., 2020c; DOI: https://doi.org/10.15482/USDA.ADC/1518706) and is separated into three datasets according to
30-year, 20-year and 10-year record lengths. To run the CLIGEN inputs, CLIGEN may be downloaded at https://www.ars.usda.gov/midwest-area/west-lafayette-in/national-soil-erosion-research/docs/wepp/cligen/.          Additional resources and materials are available at this website including the "International Conversion Programs" tool. The international CLIGEN dataset will also be added to the web interface for running the hillslope-scale erosion and runoff model, RHEM, available at https://apps.tucson.ars.ag.gov/rhem/. The station of interest will be selectable in the input parameters panel under
"Climate Station" and under "International".

## 5 Conclusions

Validation of CLIGEN inputs in the new international dataset showed reasonable agreement with parameter values for existing U.S. CLIGEN inputs. The 30-year, 20-year and 10-year datasets are generally in close agreement, although some uncertainty exists due to the assumptions taken for addressing data gaps and the degree to which short-term climate dynamics
have a role in influencing climate benchmarks. In some cases, use of higher resolution climate data for parameterization may offer an improvement over existing CLIGEN input files.

The new dataset of CLIGEN inputs allows the CLIGEN weather generator to be more readily applied to its various applications. The input files also serve to represent climate benchmarks for a selection of variables that are generally unobtainable from a single source. The coverage of stations is particularly dense in Europe, Australia, and North America, and
offers the potential to improve the spatial analysis of processes in different fields that require climate records. For a number of CLIGEN's applications, the production of climate data is a secondary concern, but is often a labor-intensive task. The use of this dataset may allow researchers to put more effort and resources towards their primary study or area of focus without needing to address the production of climate inputs.




## Appendix A

**Table A1:** *Time Pk* distribution interval values for global Köppen-Geiger climate classifications.

| Interval | 1/12 | 2/12 | 3/12 | 4/12 | 5/12 | 6/12 | 7/12 | 8/12 | 9/12 | 10/12 | 11/12 | 12/12 |
|---|---|---|---|---|---|---|---|---|---|---|---|---|
| Af | 0.22 | 0.30 | 0.36 | 0.44 | 0.50 | 0.58 | 0.63 | 0.70 | 0.77 | 0.83 | 0.90 | 1.00 |
| Am | 0.25 | 0.36 | 0.43 | 0.51 | 0.58 | 0.66 | 0.73 | 0.79 | 0.84 | 0.90 | 0.94 | 1.00 |
| Aw | 0.27 | 0.39 | 0.48 | 0.56 | 0.63 | 0.71 | 0.77 | 0.81 | 0.86 | 0.90 | 0.95 | 1.00 |
| Bwh | 0.16 | 0.26 | 0.35 | 0.43 | 0.52 | 0.61 | 0.69 | 0.76 | 0.84 | 0.90 | 0.95 | 1.00 |
| Bwk | 0.15 | 0.26 | 0.36 | 0.45 | 0.53 | 0.62 | 0.69 | 0.76 | 0.83 | 0.89 | 0.96 | 1.00 |
| BSh | 0.16 | 0.27 | 0.36 | 0.46 | 0.54 | 0.64 | 0.71 | 0.77 | 0.83 | 0.89 | 0.95 | 1.00 |
| BSk | 0.12 | 0.22 | 0.32 | 0.40 | 0.48 | 0.57 | 0.65 | 0.74 | 0.82 | 0.89 | 0.96 | 1.00 |
| Csa | 0.07 | 0.17 | 0.26 | 0.36 | 0.45 | 0.54 | 0.62 | 0.70 | 0.78 | 0.86 | 0.94 | 1.00 |
| Csb | 0.07 | 0.17 | 0.25 | 0.34 | 0.43 | 0.52 | 0.61 | 0.69 | 0.77 | 0.85 | 0.94 | 1.00 |
| Csc | 0.07 | 0.17 | 0.26 | 0.35 | 0.44 | 0.53 | 0.61 | 0.70 | 0.78 | 0.86 | 0.94 | 1.00 |
| Cwa | 0.10 | 0.20 | 0.29 | 0.38 | 0.46 | 0.55 | 0.64 | 0.72 | 0.80 | 0.87 | 0.94 | 1.00 |
| Cwb | 0.10 | 0.20 | 0.29 | 0.38 | 0.46 | 0.55 | 0.64 | 0.72 | 0.80 | 0.87 | 0.94 | 1.00 |
| Cwc | 0.10 | 0.20 | 0.29 | 0.38 | 0.46 | 0.55 | 0.64 | 0.72 | 0.80 | 0.87 | 0.94 | 1.00 |
| Cfa | 0.20 | 0.31 | 0.40 | 0.48 | 0.56 | 0.65 | 0.72 | 0.78 | 0.84 | 0.90 | 0.96 | 1.00 |
| Cfb | 0.07 | 0.15 | 0.24 | 0.32 | 0.40 | 0.51 | 0.60 | 0.69 | 0.78 | 0.86 | 0.94 | 1.00 |
| Cfc | 0.13 | 0.23 | 0.32 | 0.40 | 0.48 | 0.58 | 0.66 | 0.74 | 0.81 | 0.88 | 0.95 | 1.00 |
| Dsa | 0.17 | 0.27 | 0.37 | 0.45 | 0.53 | 0.61 | 0.68 | 0.75 | 0.82 | 0.88 | 0.94 | 1.00 |
| Dsb | 0.08 | 0.17 | 0.25 | 0.34 | 0.42 | 0.52 | 0.60 | 0.69 | 0.78 | 0.85 | 0.93 | 1.00 |
| Dsc | 0.27 | 0.38 | 0.48 | 0.56 | 0.64 | 0.70 | 0.76 | 0.81 | 0.87 | 0.91 | 0.95 | 1.00 |
| Dsd | 0.17 | 0.27 | 0.37 | 0.45 | 0.53 | 0.61 | 0.68 | 0.75 | 0.82 | 0.88 | 0.94 | 1.00 |
| Dwa | 0.16 | 0.29 | 0.40 | 0.49 | 0.58 | 0.67 | 0.74 | 0.80 | 0.86 | 0.91 | 0.96 | 1.00 |
| Dwb | 0.16 | 0.27 | 0.37 | 0.46 | 0.55 | 0.63 | 0.70 | 0.78 | 0.83 | 0.90 | 0.95 | 1.00 |
| Dwc | 0.16 | 0.28 | 0.38 | 0.48 | 0.56 | 0.65 | 0.72 | 0.79 | 0.85 | 0.91 | 0.96 | 1.00 |
| Dwd | 0.16 | 0.28 | 0.38 | 0.48 | 0.56 | 0.65 | 0.72 | 0.79 | 0.85 | 0.91 | 0.96 | 1.00 |
| Dfa | 0.15 | 0.26 | 0.35 | 0.45 | 0.53 | 0.62 | 0.70 | 0.77 | 0.84 | 0.90 | 0.96 | 1.00 |
| Dfb | 0.13 | 0.23 | 0.32 | 0.41 | 0.50 | 0.59 | 0.67 | 0.75 | 0.83 | 0.89 | 0.95 | 1.00 |
| Dfc | 0.25 | 0.36 | 0.45 | 0.53 | 0.60 | 0.67 | 0.72 | 0.79 | 0.85 | 0.90 | 0.95 | 1.00 |
| Dfd | 0.18 | 0.28 | 0.37 | 0.46 | 0.54 | 0.63 | 0.70 | 0.77 | 0.84 | 0.90 | 0.95 | 1.00 |
| ET | 0.28 | 0.41 | 0.51 | 0.58 | 0.66 | 0.74 | 0.78 | 0.82 | 0.87 | 0.91 | 0.94 | 1.00 |





| EF | 0.28 | 0.41 | 0.51 | 0.58 | 0.66 | 0.74 | 0.78 | 0.82 | 0.87 | 0.91 | 0.94 | 1.00 |

**Table A2: Statistical measures of performance. Observed (O) and predicted (P) values are compared by each metric.**

| Performance metric | Abbreviation | Equation |
|---|---|---|
| Root mean square error | RMSE | $\sqrt{\dfrac{1}{n}\sum (O - P)^2}$ |
| Percent Bias | PBIAS | $\left[\dfrac{\sum O - P}{\sum O}\right] x100$ |
| Percent Error | PERROR | $\dfrac{1}{n}\left[\sum \dfrac{abs(O - P)}{O}\right] x100$ |

**Author Contributions**

AF calculated input parameters, GA provided expertise on data management and integration with the RHEM web interface, MN and MW gave their expertise on project guidance, and all authors were involved in writing the manuscript.

**Competing Interests**

The authors declare that they have no conflict of interest.

**Acknowledgements**

The authors wish to express their appreciation for everyone involved in creating and maintaining the various climate networks that were used. Funding for this project was given through the Agricultural Research Service Headquarters Grant, and the Southwest Watershed Research Center.

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





**Table 1: Station counts for continent/region and each of the 30-year, 20-year and 10-year datasets. Oceania is the region represented by south Pacific islands and extending north to Hawaii.**

| Station Counts | North America | South America | Europe | Africa | Asia | Australia | Oceania | Antarctica | Total |
|---|---|---|---|---|---|---|---|---|---|
| 30-year | 1,860 | 170 | 2,089 | 9 | 118 | 3,423 | 4 | 0 | 7,673 |
| 20-year | 996 | 112 | 374 | 7 | 11 | 834 | 2 | 0 | 2,336 |
| 10-year | 1,332 | 8 | 413 | 6 | 52 | 864 | 19 | 0 | 2,694 |
| Total | 4,188 | 290 | 2,876 | 22 | 181 | 5,121 | 25 | 0 | 12,703 |







**Table 2: A list of CLIGEN inputs determined for each station. Parameters that require sub-daily resolutions at various frequency of measurements are denoted with "High-Res" in the Temporal Resolution column. Sub-daily resolution data was not available for all High-Res. parameters, and it is discussed how their values were estimated in some cases.**

| Variable (12 values per station) | Label | Unit | Temporal Resolution |
|---|---|---|---|
| Monthly average of daily precipitation for wet days | MEAN P | inches | Daily |
| Monthly standard deviation of daily precipitation for wet days | S DEV P | inches | Daily |
| Monthly skewness of daily precipitation for wet days | SKEW P | - | Daily |
| Monthly transition probability of a wet day given a wet day | P(W/W) | - | Daily |
| Monthly transition probability of a wet day given a dry day | P(W/D) | - | Daily |
| Monthly mean maximum 30-min precipitation intensity | MX.5P | inches/hr | High-Res. |
| Probability density function interval values of normalized time-to-peak intensity | Time Pk | - | High-Res. |
| Monthly mean of daily maximum temperatures | TMAX AV | °F | Daily |
| Monthly mean of daily minimum temperatures | TMIN AV | °F | Daily |
| Monthly standard deviation of daily maximum temperatures | SD TMAX | °F | Daily |
| Monthly standard deviation of daily minimum temperatures | SD TMIN | °F | Daily |
| Monthly mean dewpoint | DEW PT | °F | High-Res. |
| Monthly mean of daily solar radiation | SOL.RAD | langley/d | High-Res. |
| Monthly standard deviation of daily solar radiation | SD SOL | langley/d | High-Res. |
| Monthly averages of wind speed and direction | WIND (Various) | - | High-Res. |










**Table 3: The 11 predictor variables for the Gradient Boosting regression model used to temporally downscale MX.5P from GHCN-Daily data.**

| Variable | Label | Unit | Values per station |
|---|---|---|---|
| Monthly mean maximum 30-min precipitation intensity | MX.5P | mm/hr | 12 |
| Modified Fournier index | Fournier Coeff | mm | 1 |
| Monthly average of daily precipitation for wet days | MEAN P | mm | 12 |
| Monthly standard deviation of daily precipitation for wet days | S DEV P | mm | 12 |
| Monthly skewness of daily precipitation for wet days | SKEW P | - | 12 |
| Monthly transition probability of a wet day given a wet day | P(W/W) | - | 12 |
| Monthly transition probability of a wet day given a dry day | P(W/D) | - | 12 |
| Station elevation | Elev | m | 1 |
| Station latitude | Lat | deg. | 1 |
| Station coastal proximity | Coastal Prox | km | 1 |
| Calendar month (categorical variable) | Month | - | 12 |








**Table 4: Summary of the validation of parameters to the 2015 U.S. CLIGEN dataset.**

|  | 30-year dataset | | | 20-year dataset | | | 10-year dataset | | |
|---|---|---|---|---|---|---|---|---|---|
|  | RMSE | PBIAS | PERROR | RMSE | PBIAS | PERROR | RMSE | PBIAS | PERROR |
| MEAN P | 0.08 | -12.16 | 19.95 | 0.07 | 1.18 | 14.76 | 0.08 | 1.13 | 21.17 |
| S DEV P | 0.10 | -2.70 | 15.06 | 0.10 | 2.92 | 16.45 | 0.14 | 1.08 | 24.17 |
| SKEW P | 1.35 | 8.05 | 20.15 | 1.11 | 7.13 | 22.93 | 1.29 | 15.98 | 30.36 |
| P(W/W) | 0.07 | 2.48 | 10.35 | 0.06 | -1.35 | 10.33 | 0.09 | -3.68 | 16.68 |
| P(W/D) | 0.05 | -11.79 | 19.20 | 0.04 | -6.30 | 14.20 | 0.05 | -12.83 | 23.07 |
| TMAX AV | 3.49 | 3.18 | 3.97 | 5.43 | -0.41 | 6.77 | 3.75 | 0.66 | 4.28 |
| TMIN AV | 4.56 | -8.55 | 15.79 | 6.23 | -10.62 | 13.67 | 4.76 | -7.93 | 11.33 |
| SD TMAX | 1.07 | 7.93 | 9.01 | 1.37 | 11.56 | 13.28 | 1.30 | 9.62 | 11.85 |
| SD TMIN | 1.53 | 6.87 | 11.34 | 1.22 | 7.80 | 13.01 | 1.04 | 4.45 | 10.98 |
| SOL.RAD | 22.55 | -1.08 | 5.85 | 29.10 | -2.90 | 5.87 | 26.91 | -2.75 | 5.65 |
| SD SOL | 51.85 | -135.54 | 146.33 | 68.09 | -193.42 | 202.42 | 63.04 | -173.21 | 181.51 |
| MX .5 P | 0.23 | 24.93 | 29.95 | 0.27 | 28.37 | 31.93 | 0.31 | 33.26 | 37.30 |
| DEW PT | 3.66 | 5.62 | 8.94 | 2.00 | 0.45 | 5.14 | 2.56 | 0.48 | 5.85 |
| Time Pk | 0.33 | 30.92 | 33.43 | 0.30 | 28.33 | 31.08 | 0.30 | 28.77 | 31.66 |








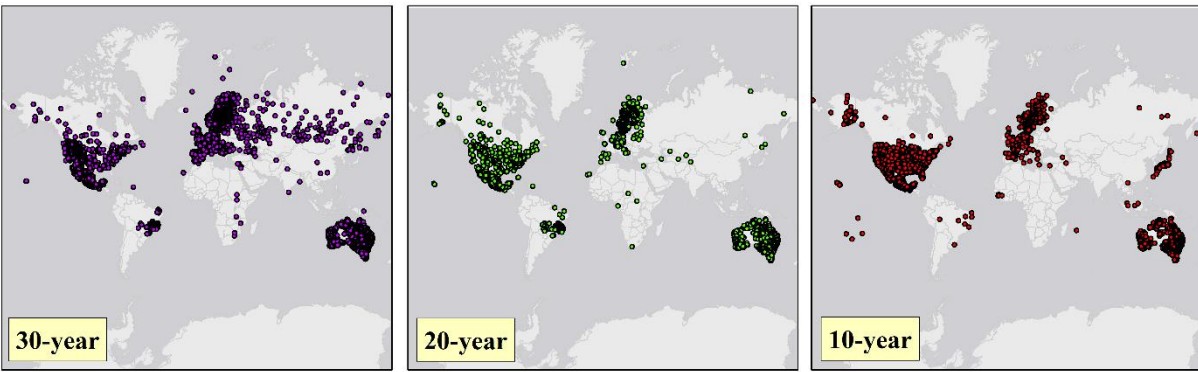

**Figure 1: Coverage of the three international CLIGEN input datasets according to the record length used to produce the monthly input parameters. The locations correspond to those of the GHCN-Daily stations accepted for use.**