# Peer review of "Climate benchmarks and input parameters representing locations in 68 countries for a stochastic weather generator, CLIGEN"

_Earth System Science Data, 2020_

## Referee Comment (RC1) · Anonymous Referee #1 · 13 Oct 2020

The paper presents a serious attempt to develop and share a database for parameter values for CLIGEN so that CLIGEN can be used more widely around the world.

While data sources for individual weather variables were described in reasonable details. What is missing is the table summarizing the spatial resolution for each.

Based on my limited knowledge of various data sets, some, like precipitation, are clearly site-specific, while others are grid-based, interpolated, like temperature. How were temperature parameter values prepared for individual sites? Were gridded temp data mapped to individual precipitation sites through spatial interpretation techniques? The same applies to solar radiation data.

[Figure]

Recognition of the issue with record length is useful, but not critical. In CLIGEN parameter file, the number of years of data is recorded. I would leave at that, Caveat Emptor!

(PS, I have had a close look at the parameter files generated. For 20-year data set. The years used are still 30.

As an example, randomly,

" LATT= -26.27 LONG= 117.78 YEARS= 30. TYPE= 3"

)

Fig.1 Why are there more sites with 30-year data than those with 10? Any station with 30 year would also have 10 years of data?

Presentation is good, manuscript readable. All the equations require close attention though:

(1) 'n days' 'MEAN P' should not be used as variables in the equation. Equations need to be readable, clear, precise.

(2) The equation is wrong, once the summation sign is used. there is no need for all other terms. That is what the summation is for.

(3) Again 'Time Pk(i)', any variables with a space ' ' in them can lead to confusion

One bracket is missing from the equation in the third row in Table A2.

If there is space in the variable name, use ' ' for the variable.

---

## Referee Comment (RC2) · Anonymous Referee #2 · 19 Oct 2020

In reviewing "Climate benchmarks and input parameters representing locations in 68 countries for a stochastic weather generator, CLIGEN" I found an usefull dataset for calibration of CLIGEN parameters. The proposed results may support further applications not focused on climate data but that use them, as the authors state.

Data are available at the exposed link and easily accessible and usable, the manuscript is readable.

In lines 83-84 and 115-118 slightly mismatching statement are proposed. Using complete months in non continuous series could drive to incompatibilities in temporal comparison of the proposed parameters?

[Figure]

Overplotting occurs in figure 1, maybe a thematic raster "distance from nearest location" can enhance the information provided? Furthermore details on spatial coverage of the proposed parameters could be provided.

The collection and harmonization of international climate data encounters notorious obstacles, their once for all overcome falls in the goals of this work. The methods used are soundly reported, but -I miss an explicit criterion for gridded model temperature values: radiation is in ERA5 dataset. Why use gridded temperature and not radiation? -About SD RAD, the 2.6 closes on GLDAS-Ameriflux comparison, the proposed global parameters rely on the continue model of GLDAS 3h values?

I would expect 30y points to be fulfilling the requirements of 10y ones, it doesn't look so in figure 1. May the definition of time-series at line 11 be improved using "maximum" or "available" rather than "minimum"?

Somewhere in the text CLIGEN parameters are referred with no introduction (ie. lines 92, 236), their presentation would get the text easier to read.

―――――――――――――――

---

## Author Comment (AC1) · 24 Nov 2020

Summary of Changes: Revisions made to address the referee comments improved the clarity and accuracy of the paper. A response is given below to each comment where a response was required. Minimal changes were made to the paper beyond what was done to address the comments. A revised version of the dataset is now available at Ag Data Commons. The revision to the dataset corrected two metadata issues: first, as the referees point out, the record length (10, 20 or 30 years) was not correctly shown in the headings of the 10 and 20-year .par files, and this is now corrected; second, improvements were made to the formatting of locality names in the headings of the .par

files. There were also changes made to correct inconsistencies in MX.5P and transition probability parameter values for very dry climates or months. Some values were predicted be smaller than 10-3 (or two decimal places), which is the minimum value allowed by CLIGEN, and were therefore set to zero. This sometimes corresponded with non-zero MEAN P values. If this was the case, the MX.5P or transition probability was set to the minimum allowed value of 10-3. This had a small effect on the validation metrics in Table 4.

RC1 Key Points:

RC1 #1) While data sources for individual weather variables were described in reasonable details. What is missing is the table summarizing the spatial resolution for each.

Since the precipitation parameters were point-scale (or site-specific), spatial resolution is relevant only for the temperature and solar radiation parameters derived from gridded products. The climate models that were used, ERA and GLDAS, both have 0.25° x 0.25° resolution, and this is now stated in the text instead of in a table.

RC1 #2) How were temperature parameter values prepared for individual sites? Were gridded temp data mapped to individual precipitation sites through spatial interpretation techniques? The same applies to solar radiation data.

This relates to the previous comment. In the same revision to include the spatial resolution of the climate models, it is now stated that no weighting of values based on proximity of a station to neighboring cells, or other forms of interpolation, is done. The 0.25° resolution translates to ~28 km resolution at the equator, which is reasonable resolution for this application. Using the original values from the models also enables easier interpretation of existing uncertainty information for respective models.

RC1 #3) Recognition of the issue with record length is useful, but not critical. In CLIGEN parameter file, the number of years of data is recorded. I would leave at that,

Caveat Emptor!

We feel that the record length issue is worth explaining in the text because it may complicate the interpretation of the data and any future validation that is done through comparison to other climate records, particularly for non-stationary climates or climates with long-term cycles.

RC1 #4) I have had a close look at the parameter files generated. For 20-year data set. The years used are still 30.

This metadata issue is now corrected in the revised dataset.

RC1 #5) Fig.1 Why are there more sites with 30-year data than those with 10? Any station with 30 year would also have 10 years of data?

Correct, any station with 30 years of data would be viable for the 10-year dataset. The longest possible record length (of 10, 20, or 30 years) was used for a given site, such that if a 30-year dataset was possible, a 10 and 20-year dataset were not made in addition. So, no site had multiple datasets created for it. This is now stated in the text, and this partly explains why the 30-year dataset has the most locations. It is also the case that many NOAA-GHCN sites included a long backlog of data at the time of being added to the network, for which 30-year datasets were possible, while the 10-year datasets tended to come from newer installations without a long backlog.

RC1 #6) (Equation) (1) 'n days' 'MEAN P' should not be used as variables in the equation. Equations need to be readable, clear, precise.

In the context of the equation, variable names for n days and MEAN P were changed to shorter names with no spaces, making the equation easier to read. The identities of the new variable names are explained in the text under the equation.

RC1 #7) (Equation) (2) The equation is wrong, once the summation sign is used. there is no need for all other terms. That is what the summation is for.

[Figure]

Correct, the "+...+" inside of the summation operator should not be shown. This was meant to be ",...," which clarifies what the set of terms is being summed.

RC1 #8) (Equation) (3) Again 'Time Pk(i)', any variables with a space ' ' in them can lead to confusion One bracket is missing from the equation in the third row in Table A2. If there is space in the variable name, use ' ' for the variable.

The space was removed from the Time Pk variable name in the equation and where it is used in sections of the text. The space was previously used to be consistent with what is shown in CLIGEN .par files, but the connection between the two labels is evident and shouldn't lead to confusion. Several of the CLIGEN parameter names have spaces in them but are not used frequently in the text like Time Pk is. So, spaces in the other names were kept. Also, the percent bias equation was missing a bracket around (O – P), which is now fixed.

RC2 Key Points:

RC2 #1) In lines 83-84 and 115-118 slightly mismatching statement are proposed. Using complete months in non continuous series could drive to incompatibilities in temporal comparison of the proposed parameters?

The statement on 83-84 does imply continuous records should be used. The handling of the pervasive data gaps in NOAA-GHCN records becomes a source of error, and complicates validation using other datasets with the same temporal ranges. A stronger statement is made about the uncertainty from using non-continuous records with mention of the uncertainty associated with non-stationary climates, long-term climate cycles, and the complication that arises when comparing to other climate data.

RC2 #2) Overplotting occurs in figure 1, maybe a thematic raster "distance from nearest location" can enhance the information provided? Furthermore details on spatial coverage of the proposed parameters could be provided.

A new figure (Fig. 2) was added that shows a raster of station density. This allows the

reader to see the relative density of stations in places where overcrowding makes this impossible.

RC2 #3) The collection and harmonization of international climate data encounters notorious obstacles, their once for all overcome falls in the goals of this work. The methods used are soundly reported, but -I miss an explicit criterion for gridded model temperature values: radiation is in ERA5 dataset. Why use gridded temperature and not radiation? -About SD RAD, the 2.6 closes on GLDAS-Ameriflux comparison, the proposed global parameters rely on the continue model of GLDAS 3h values?

The temperature and solar radiation parameters come from ERA and GLDAS gridded climate models, respectively, and this is now made more explicit in the text. A lack of clarity about this may have been leading to a misunderstanding originating from the discussion of the use of point-scale Ameriflux ground observations for validation that solar radiation parameters did not come from a gridded product. So, answering the question in the comment, it is correct that the parameters rely on the GLDAS 3h values.

RC2 #4) I would expect 30y points to be fulfilling the requirements of 10y ones, it doesn't look so in figure 1.

The same question was addressed in response to RC1 #5.

RC2 #5) May the definition of time-series at line 11 be improved using "maximum" or "available" rather than "minimum"?

"Minimum" was changed to "available". Use of "available" in the context of record length may suggest that some screening or filtering was done, which is the case.

RC2 #6) Somewhere in the text CLIGEN parameters are referred with no introduction (ie. lines 92, 236), their presentation would get the text easier to read.

Table 2 gives all parameter definitions, so rather than restating their definitions in the text, reference to Table 2 is made at the beginning of each of the major methods sections where the respective parameters being calculated are generally listed.

[Figure]

---

## Author Response (AR2)

**Summary of changes**:

No additional changes were made beyond what was needed to address the editor's suggestion below, and no additional typos were found.

**Comment:**

Small residual concern related to authors' use of Google Earth Engine. We understand ease of access and other convenience, but we also know that Google changes (updates) products on unpredictable and undocumented time schedules. Please can authors add a small table, perhaps Appendix Table 3, that lists exact original sources, versions, access dates of those GEE products used here? Such information allows users to know exactly what information informed these data products but also to expect and identify changes.

**Response:**

An appendix table (Table A3) was added. The table contains metadata for the four different climate models that were accessed using Google Earth Engine. This includes links to the Google Earth Engine Catalog for each product, version numbers, date of access, and the original source/organization that produced the data. In-text references to Table A3 were added in sections 2.4 and 2.5 where the use of the climate models is discussed.